# Sustaining or Declining Physical Activity: Reports from an Ethnically Diverse Sample of Older Adults

**DOI:** 10.3390/geriatrics6020057

**Published:** 2021-06-03

**Authors:** Ruth Tappen, Edgar Ramos Vieira, Sareen S. Gropper, David Newman, Cassandre Horne

**Affiliations:** 1Christine E. Lynn College of Nursing, Florida Atlantic University, 777 Glades Road NU-84, Boca Raton, FL 33431, USA; sgropper@health.fau.edu (S.S.G.); dnewma14@health.fau.edu (D.N.); cexantu3@health.fau.edu (C.H.); 2Nicole Wertheim College of Nursing & Health Sciences, Florida International University, 11200 SW 8th St, AHC3-430, Miami, FL 33199, USA; evieira@fiu.edu

**Keywords:** physical activity, sustaining activity, older adults, disparity, purpose-driven, health benefits

## Abstract

Over 80% of adults in the US fail to meet the ≥150 min weekly physical activity guideline; 40% age ≥ 75 are entirely inactive. The study purpose was to understand the reasons why community-dwelling older adults (age ≥ 60) from diverse backgrounds increase, sustain, or decline in their physical activity levels over time. Sixty-two older adults were interviewed. Two-thirds of the African Americans, 57% of the Afro-Caribbeans, and 50% of the European Americans reported being less active than 2–3 years ago. Reasons for activity decline included health issues (e.g., pain, shortness of breath), lack of time, interest, or motivation. Reasons for sustaining or increasing activity levels included meeting personal goals, having a purpose for remaining active, or feeling better when active (e.g., it is important to keep moving, good for the joints, going on a cruise). Themes identified were pride in maintaining activity, goal-driven activity, pushing oneself to get past pain or fatigue, and some confusion between social and physical activity in participant reports. The results indicate widespread acceptance that activity is beneficial, but that knowledge alone was insufficient to maintain activity levels over time unless individuals had a goal or purpose (“means to an end”) and could overcome their physical and psychological barriers to physical activity.

## 1. Introduction

The demonstrated health benefits of regular physical activity for older adults include lower risk of premature morbidity and mortality. In particular, physical activity can contribute to improved management of chronic conditions such as type 2 diabetes, cardiovascular disease, falls risk, and depression and it can improve quality of life [1]. Most older adults believe that continuing to be physically active would be beneficial to their health, yet over 80% of all adults in the U.S. fail to meet the World Health Organization physical activity guidelines that encourage at least 150–300 min of physical activity [2]. Twenty-eight percent of adults age 50 and older are entirely inactive; this proportion rises to 40% at age 75 and above. Considerable interindividual differences in activity levels are found within the older adult population. Significant disparities in physical activity levels by sex, race, ethnicity, education, income, disability, and place of residence have also been identified [3].

An abundance of research has been done examining factors associated with participation in physical activity among older adults [4,5,6,7,8]. Social support from family, increased psychological well-being, and higher overall expectations of aging, among others, have been positively associated with increased activity [4,5,6]. In contrast, barriers such as pain, lack of interest, depression or anxiety, poor health status, and lack of access to a gym or facility offering classes have been identified as barriers that reduce or limit physical activity among older adults [5,7,8]. In a recent systematic review, a combination of individual attitudes (e.g., motivation and self-efficacy) and social determinants (gender, education) were found to influence physical activity levels in older adults [9]. Cavazzotto et al. identified the effects of social-ecological factors related to intrapersonal (for example, age) and environmental (climate, for example) characteristics on regular physical activity in adults aged 46 to 60 [9]. While personal and climate/weather characteristics had stronger effects than the built and sociopolitical environments, the authors highlighted the multifactorial nature of influences on regular physical activity practice [9]. The findings of these studies have been critical in the development of appropriate intervention strategies to encourage the initiation of physical activity. On the other hand, little is known about factors that are associated with increasing, sustaining, or declining physical activity levels over time, which is necessary to achieve and maintain the long-term benefits of physical activity.

The purpose of this study, then, was to examine older adults’ reports of sustaining or declining physical activity levels over time and their explanations of these trends in a diverse, community-based sample of older adults using a predominantly qualitative mixed-methods approach.

## 2. Materials and Methods

### 2.1. Design

An explanatory sequential mixed-methods design was employed for this study. In this design, the research begins with a quantitative phase followed by a qualitative phase in which further understanding of a particular quantitative result is explored in depth qualitatively [10]. For the initial quantitative phase, data was obtained from the Healthy Aging Research Initiative (HARI). The HARI database is a large, diverse sample of older adults living independently in three metropolitan counties of Southeast Florida aged ≥60 who are cognitively unimpaired (MMSE score ≥ 23) and able to walk 20 feet independently or with the aid of a cane or walker [11]. It was evident from a review of the HARI database that there was inadequate information on participants’ physical activity, an essential element of a healthy lifestyle. To address this, we conducted telephone interviews with a subsample of the English-speaking portion of the larger HARI sample. Quantitative information about these 62 individuals was drawn from the Phase 1 HARI survey and qualitative information from the Phase II telephone interviews. The in-depth qualitative data were obtained from the follow-up telephone interviews.

### 2.2. Data Collection

#### 2.2.1. Survey Data

During the initial HARI survey, a wide range of sociodemographic, health history, social network, and psychological state questions were asked of participants. Data on participants’ age, gender, ethnic group membership, and receipt of Medicaid (health care coverage for those with very low incomes as a proxy for income level) were obtained during collection of the survey data. Cognitive function was measured by the Mini Mental State Examination [11]. The MMSE test is a 30-point questionnaire that is used extensively in clinical and research settings to measure cognitive impairment and commonly used to screen for dementia [12]. Function was measured by the FAQ (Functional Activities Questionnaire), which is a measure of IADLs (Independent Activities of Daily Living) intended for use with older adults in a community setting [13]. These measures were administered by trained assistants.

#### 2.2.2. Interview Data

Sixty-two older adults from the HARI database were asked open-ended questions about their physical activity in follow-up phone calls 3 to 4 months after the initial HARI data collection. They were asked if their activity had increased, declined, or stayed the same and the reasons why:Have you made any change in your level of physical activity in the last several (2–3) years?Do you think you should be more physically active than you are? Why or why not?What, if anything, makes it difficult to make or sustain a change in activity level?

Responses were transcribed and the transcriptions checked for accuracy prior to analysis.

#### 2.2.3. Protection of Human Subject

This study was reviewed and approved by the University Committee for the Protection of Human Subjects. The study was explained to the potential participant and written informed consent was provided by the participant prior to engaging in study activities.

### 2.3. Data Analysis

#### 2.3.1. Quantitative Analysis

Analysis of the quantitative data on characteristics of participants drawn from the HARI database included generation of frequencies and percentages followed by comparisons across groups, particularly by ethnic group. This was done utilizing one-way ANOVA (SAS 9.4) for continuous variables and chi square testing for categorical variables. Fisher’s exact test was used where one or more cell frequencies were less than 5 for which the phi coefficient was reported [14]. Post-hoc comparisons were conducted on those characteristics that differed significantly across ethnic groups. A binary logistic regression of ethnic differences across physical activity trends (increased, decreased, or sustained activity) controlling for age and BMI was also conducted.

#### 2.3.2. Qualitative Analysis

Analysis of the interview data began with reading of the entire set of transcribed interviews to develop a general understanding of participants’ responses and create a codebook to guide the first level of coding. This first level of coding was both descriptive and concrete with minimal interpretation of the responses. Coders were blinded to ethnic group membership to minimize assumptions coders might have had regarding differences across ethnic groups. Differences in coding across the three coders were reconciled and consensus reached. The second level of analysis was interpretative in nature. At this level, the coders as a group discussed participant responses to understand the meaning underlying their statements. These interpretive level themes were reviewed, discussed, and agreed upon by all members of the research team [10,15,16]. Reports of increased, sustained, and decreased activity in the last 2–3 years were coded by the investigators and the results (increased, decreased, sustained) were transformed into a nominal level quantitative variable for inclusion in the quantitative analysis [15]. Negative case analysis was also done. This approach considers alternative explanations and instances in which participants did not express the prevailing viewpoint [17]. As a result, we not only reported the facilitators and barriers to sustaining physical activity level but also what participants did not say that is generally expected to be an important facilitator or barrier to sustaining activity.

## 3. Results

### 3.1. Sample Characteristics

Of the 62 older adults interviewed, *n* = 21 (34%) were self-identified as African Americans (AA), *n* = 24 (39%) as European Americans (EA) and *n* = 17 (27%) as Afro-Caribbeans (AC). Twenty (32%) were male; *n* = 42 were female (68%). Mean 3.3.age was 76 (SD 8.06) with a range of 60 to 94. Years of education averaged 14 (SD 4.85) with a range of 7 to 27. Years having lived in the United States averaged 62 (SD 21.06) with a range of 13 to 94 years, reflecting the immigrant status of the majority of the Afro-Caribbean participants. Ten participants (16%) were receiving Medicaid benefits, *n* = 52 (84%) were not. All were fluent in English.

### 3.2. Comparison by Ethnic Groups

#### Ethnic Group Differences in Relevant Characteristics

There were some differences across ethnic groups in the sociodemographic characteristics of the participants (Table 1). Although the groups did not differ significantly by age, there were significant differences across groups in terms of years of education, gender distribution, and years living in the U.S. There were significantly more males in the European American group (58%) compared with 10% in the African American group. The mean years of education were also significantly higher in the European American group (17.78; SD 4.65) than in the African American group with a mean of 11.80 (SD 3.04) or Afro-Caribbean group with a mean of 10.06 (SD 3.23). Reflecting the immigrant status of the Afro-Caribbean group, only 2 of these 17 participants were born in the U.S. Similarly, years having lived in the U.S. differed significantly across groups.

There were also differences by ethnic group on several additional characteristics. While the groups did not differ by cognitive function, ability to carry out independent activities of daily living, or reported change in activity level over the last 2–3 years, there was a significant difference in BMI based on measured (not reported) weight and height with the African American participants having significantly higher BMI at a mean of 43.29 (SD 6.56) compared to 37.66 (SD 6.52) for Afro-Caribbean participants and European American (35.22; SD 3.81) (Table 1).

### 3.3. Sustaining Physical Activity

In response to the question about recent change in physical activity, 34 (55%) reported that they were less physically active than 2–3 years ago, 23 (37%) reported the same activity level, and 5 (8%) reported an increase in physical activity. Sixty-seven percent of the African Americans (*n* = 14), 53% of the Afro-Caribbeans (*n* = 9) and 46% (*n* = 11) of the European Americans reported being less physically active currently than they were 2–3 years ago (See Figure 1). This difference was not significant across ethnic groups, *p* = 0.21. A greater percent of African American participants reported a decline in activity and a smaller percent remained the same but actually had a higher percent (although still small) reporting an increase. There were some similarities between the Afro-Caribbean and European American reports: fewer declined, more remained the same and fewer reported an increase than in the African American group (See Figure 1).

We combined two of the physical activity categories, sustained and increased, due to the limited number of those that reported an increase. Since there were statistically significant differences in BMI and age across the ethnic groups, as indicated in Table 1, a follow-up binary logistic regression was conducted to investigate ethnic differences on self-reported physical activity decline, and as can be seen in Table 2 there was no statistically significant difference in African American and Afro-Caribbean changes in physical activity levels when controlling for BMI and age *p* = 0.122, and *p* = 0.088, respectively.

### 3.4. Explaining Activity Levels

Five distinct explanations for current activity levels were identified based upon participant reports: participants who reported they were physically active and had positive feelings about their activity; those who were struggling to remain physically active; participants who expressed a wish to do more; those who were inactive due to one or more physical barriers; and those who were inactive for lack of interest, motivation, and other deterrents such as time and ability to find and/or access a group exercise program.

Positive on Physical Activity

This segment of the interviewees described engaging in regular physical activity, many specifying the type of activity as well.

I think I am doing a good job of regular exercise. (AC)Stooping, bending, stretching for a few minutes a day; I do a lot of walking and we do dancing at church. (AA)

Many added that being physically active made them feel good and that it was beneficial health-wise:I always feel better when I exercise. (AC)The body is meant to move. (You) can’t just sit. (EA)I know it’s important to keep moving. (AC)Exercise is good for you. It’s important to move your joints if you have arthritis. (EA)More physical activity keeps your body going, which keeps your brain going. (AA)

2.Struggling to Sustain Physical Activity.

A second segment within those reporting being physically active were the participants who reported making an effort to be active but experiencing some difficulty with sustaining physical activity.

Sometimes my hip hurts and I don’t feel like walking. (AC)When I exercise, there is pain in my neck and back due to arthritis. (EA)I force myself to walk each day. My long-distance walking is much shorter now because I get short of breath. (AA)

3.Wishing They Could Do More.

A smaller but also important segment were those who expressed dissatisfaction with their current level of activity and a desire to do more.

If I had known I would live this long, I would have taken better care of myself. I want to think about how to improve it and make it better. (AA)Want to be more active. Back pain and arthritis. (EA)I think I should get involved and exercise. I know I should exercise for my bad knees. (AA)My metabolism is slow; everything I eat turns to fat. I would love to be more active. (EA)I know it’s good for me and I should be doing it more often. (AC)

4.Physical Barriers Impeding Activity.

Among those who reported they were inactive, a substantial segment indicated that they faced significant physical barriers, sometimes multiple barriers:Poor circulation, varicose veins, and swelling. (AC)The pain in my knees makes it difficult to do much walking. (AC)No exercise anymore because of shoulder problems. (AC)I don’t have the energy. I get tired so fast. (AA)I get short of breath. (AA)

5.Psychological Barriers Impeding Activity.

This final segment of the participants who were interviewed provided a wide range of reasons why they were inactive that were not related to specific physical barriers, but primarily to little motivation to exercise. One of the more specific reasons was that the participant was still employed and had little time or energy for physical activity.

Would go to the gym more if I wasn’t working part-time. (AC)Finding time after work, when I still have energy. (AA)Not enough time. (EA)I have too much to do. (AC)

However, many more participants attributed their inactivity to lack of motivation, laziness (their term), feeling down, or sleeping too much.

I am lazy and do what I want. (AA)Probably I could do more, but I just don’t. (AC)Lack of motivation. (AC)I’m not doing anything right now. (AC)By sleeping so many hours, I have less time to be active, have gained some weight. (EA)

Finally, there was a small group that expressed no interest in becoming physically active or exercising.

I don’t want to become more active. (AC)I don’t like exercising. I can’t find more places to dance. (EA)

### 3.5. Themes

A number of themes were identified across these participant reports. These themes included a strong sense of pride among those who continue to be physically active; a goal, purpose, or “means to end” expressed either directly or indirectly; and the concerted effort “forcing myself” of some trying to sustain their physical activity, but few if any arguments that a participant was “too old” to exercise and little indication of cultural/ethnic differences across the articulated deterrents and facilitators of exercise [18]. It was also noted that some participants confused social activity with physical activity, mentioning their volunteer efforts or church involvement (other than “dancing at church”) as evidence of their activity.

*Pride* was expressed during the interviews by those who were physically active. This pride was reflected in reports of how well the participant was doing in regard to being physically active or simply in the recitation of the various activities in which they engaged over a week.

A *goal* or *purpose* for exercising was frequently mentioned. Some were specific (getting ready to go on a cruise, playing with grandchildren), others were more general, directed toward feeling good, staying healthy, maintaining function, and staying independent. This is what Morgan and colleagues termed activity as a “means to an end” [18].

*Forcing oneself.* The effort some participants had to make to sustain their physical activity level was evident in the struggling group. Some said they had to force themselves to keep moving.

Several themes were derived from negative case analysis. In other words, some responses that might have been expected were not articulated by the participants.

### 3.6. Too Old

While a significant segment of the sample expressed little or no interest in becoming physically active, none declared that it was not good for someone their age or that there would be no benefit to becoming active. There was no negativity expressed toward exercise in the later years or other reasons given why an older adult should not be physically active.

### 3.7. Influence of Friends and Family

Interestingly, there was virtually no reference to friends, family, or community encouragement to remain active. However, some participants who sustained physical activity mentioned group environments, such as church, the YMCA, and gyms, where they engaged in physical activity.

### 3.8. Effect of Ethnic Group Membership

There was an expectation in the research team that ethnic group membership would be an influence on efforts to sustain physical activity. This was the reason for blinding coders to ethnic group membership until the coding was completed. While they noted that the Afro-Caribbean participants seemed to be the most talkative (i.e., disposed to engage in conversation) after unblinding, they did not find substantial differences in reasons for increasing, sustaining, or declining physical activity. There was, however, a potential trend noted in the greater number of African Americans reporting a decline in activity and their significantly higher BMI levels (noted in Table 1 and Figure 1).

### 3.9. Confusion between Physical and Social Activity

*Confusion between physical and social activity* was clear in a number of participant responses describing social, community, or religious pursuits as their activity. What was not clear, however, was the extent to which this was conceptual confusion on their part or unclear communication in the question asked due to use of the term “activity” instead of “exercise”.

## 4. Discussion

### 4.1. Summary of Results

Of the 62 African American, European American, and Afro-Caribbean older adults interviewed, 34 (55%) reported that they were less physically active than they had been two to three years ago, 23 (37%) reported the same activity level, and a very small number, only 5, (8%) reported an increase in physical activity. The reasons behind these trends in activity level provided by participants were a positive stance on being physically active, struggling to sustain physical activity, wishing they could be more active, physical barriers to activity, and psychological deterrents to physical activity. The themes identified were a strong sense of pride related to sustaining physical activity, having a goal or purpose related to sustaining activity, and the considerable effort some expended to remain active. On the negative case side, i.e., not noted in participant responses, there were few arguments that the participant was too old to exercise, little difference in reasons reported for sustaining or declining physical activity evident across ethnic groups, less mention of the social aspects of physical activity than was expected, and some confusion among participants concerning the concept of social activity versus physical activity.

### 4.2. Toward a Lifetime of Physical Activity

The importance of physical activity is well recognized by both science and the public. The CDC *Physical Activity Guidelines for Americans* recommend at least 150 min of moderate-intensity or 75 min of vigorous-intensity physical activity per week [19]. Most exercise programs for older adults focus on achieving these goals for the duration of the program (3 to 6 months typically). However, physical activity is a lifelong goal and people need to sustain their levels of activity their entire life to improve overall health and reduce the risk and burden of chronic diseases. More focus on initiatives to sustain these levels for longer periods (year after year for decades) is needed. Such programs would need to involve strategies to help people integrate physical activity into their lifestyles. Ideally, such programs and educational initiatives would begin in childhood. In our experience with exercise intervention studies with older adults, the strongest predictor of those who are likely to continue and complete the programs are those who say they used to exercise when they were younger [20]. In the current study, one participant unhappy with current activity levels expressed this sentiment “If I had known I would live this long, I would have taken better care of myself.” Providing younger adults with positive expectations of activity in aging could be useful in sustaining engagement in physical activity over a lifetime.

### 4.3. Facilitators and Barriers to Sustaining Activity

Many of the individuals in this study who were sustaining their physical activity levels exhibited similar characteristics, including motivation generated by various perceived benefits, pride in their continued activity, and reporting a personal, meaningful goal or goals. Goal-directed behavior has been linked positively with physical activity among older adults in other studies [8,21]. Examples of specific personal goals provided by participants included being able to play with grandchildren or going on a cruise. More general goals were to live a healthy life and remain independent. Such positive attitudes toward physical activity are also consistent with theory-based constructs such as the theory of planned behavior (TPB), social cognitive theory, and the health belief model, a few of the many theories that are often employed to explain health-related behaviors and to design interventions to engage older adults in physical activity [22,23,24,25,26,27].

Individuals struggling to sustain physical activity, those wishing they could do more, and those with existing physical barriers to activity often mentioned pain and fatigue related to existing conditions. Pain is a common barrier to participating in and sustaining physical activity [7]. According to the TPB, perceived barriers influence an individual’s decision to perform a specific behavior (physical activity). Notably, however, some of the older adults worked past barriers such as pain, while others did not. As indicated by our findings, some older adults believe that moving more and engaging in physical activity would increase their pain. This is a misconception. Physical activity has been found to decrease pain. Older adults with less sedentary time and greater levels of daily light physical activity have greater pain inhibitory capacity [28]. A review of the use of physical activity to manage pain in older adults found that most studies indicate decreased pain after exercise training [29]. Knee pain was a common barrier for exercise and a common reason given to reduce levels of physical activity by the participants. However, a systematic review of 49 studies investigating if long-term physical activity was safe for older adults with knee pain found that low impact exercises not only did not result in significant increases in pain but also reduced the risk of needing a total knee replacement [30]. Tailoring the physical activity of the individual to minimize pain can help the individual both overcome the barrier and build self-esteem and self-efficacy [18]. Guidance from professionals who may be able to clear up misconceptions about the relationship of exercise and pain may contribute to sustaining activity [31].

Older adults who have led sedentary lifestyles are also at risk for sarcopenia, characterized by reductions in muscle mass, strength, and function. The loss of lean body mass resulting from sedentary behaviors can be quite large. For example, Kortebein et al. reported lean mass losses of 1500 g, of which 950 g (63%) came from muscle in the lower extremities, in a group of 12 healthy older adults when comparing values pre- and post-10 days of bedrest [32]. These changes in muscle mass should be recognized and guidance in increasing lean muscle mass through diet and exercise should be provided to sedentary older adults to encourage initiating and sustaining physical activity.

Self-efficacy, i.e., confidence in performing an action, has been linked to physical activity. According to Bandura (the social cognitive theory), and within the context of physical activity, self-efficacy influences whether the individual tries an activity, persists in the physical activity through encountered difficulties, and finally succeeds or fails [27]. Low self-efficacy has been associated with physical inactivity [8]. Interestingly, few indications of self-efficacy were apparent in responses given by the participants engaging in and sustaining physical activity in this study. One exception was concern about falling as a barrier to sustaining activity levels reported by several participants. Otherwise, individuals wishing to do more and those with expressed physical and psychological barriers to activity did not implicate low self-efficacy. It also is unclear whether the documented sustaining of physical activity contributed to the strong self-esteem or vice versa. Self-esteem was clearly present in the pride expressed by a number of participants who were physically active. Both self-efficacy and physical activity have been positively linked with self-esteem in previous research [33].

### 4.4. Tailoring Physical Activity Interventions

The multifactorial nature of reasons for increasing, sustaining, or declining levels of physical activity calls for reframing our traditional approach to helping older adults initiate and maintain participation in physical activity and adopting a multipronged approach that is tailored to the individual. Our results suggest that people who understand and value remaining physically active are more likely to make a conscious effort to sustain their activity levels at an older age despite barriers such as pain and declining physical function. The results also suggest that others need guidance as to how they can continue being active despite pain, fatigue, and other physical and emotional barriers and assistance in identifying meaningful personal goals for sustaining physical activity. It was clear that a number of participants could have benefited from some information and assistance in engaging activities that can reduce pain and fatigue and decrease fall risk. Goal-setting can promote physical activity, especially when the established goals provide a physical or emotional benefit. More effective goals are ones that are of personal importance (i.e., meaningful) to the older adult and directed toward more than general health benefits [18,34,35]. In a review of interventions to address the effects of social determinants on health in ethnic minority groups, Pool and colleagues found that the effective interventions had a theoretical base and used a group format. These findings also need to be considered in designing interventions to sustain physical activity in older adults [36].

The confusion of social and physical activity evident in some participants’ responses suggests care needs to be taken in crafting questions asked of older adults during primary care visits and similar situations. Instead of asking “Are you staying active?” ask “Tell me how physically active you are: what kinds of physical activities do you do? How often do you do them in a week?”

In addition to person-centered, meaningful goal setting, and addressing personal barriers, each individual’s personal preferences and interests should be recognized. For example, providing targeted activities (such as dancing, yoga, exercise classes, working out in a gym or with video at home, or simply walking in one’s neighborhood, the hallways of senior residence, or a park) that the individual is predisposed to undertake can help to reframe the physical activity from “needed for health” to “enjoyable” [21]. The inclusion of selected music or visual images during the activity can provide added enjoyment and be of emotional benefit [34]. For older adults who are lonely or bored, promoting physical activity as a source of adding structure to the day (creation of a routine) and increasing social contact with others can contribute to a sense of satisfaction, mastery, and feeling of purpose [18]. The promotion of such feelings, as well as boosting self-esteem, strengthening self-identity, and role fulfillment through physical activity, represent some of the many steps that are needed to “reframe” approaches to sustaining and increasing physical activity.

### 4.5. Limitations

The representativeness of a sample and the responses obtained are always issues in qualitative research. This is counterbalanced somewhat by the richness of the information obtained. Nevertheless, the responses obtained reflect the perspectives of the individuals interviewed but are not necessarily representative of other individuals of the same age, gender, or ethnic group. Furthermore, there were some significant differences in the sociodemographic characteristics of the ethnic groups represented in our sample. Despite these differences and the blinding of the coders, we found remarkable consistency in responses across the ethnic groups. We would not conclude from this that there are, therefore, no differences to be found but that additional, more in-depth questions need to be asked and then analyzed by ethnic group and that differences may be more subtle and nuanced than anticipated and more difficult to capture without in-depth interviewing than originally expected. Additionally, the study was underpowered to detect any differences by ethnic group membership in the proportions who had sustained their level of activity versus declined over the last 2–3 years. The same may be true of the apparently minimal effects of social groups, i.e., that their influence is subtle, and participants may have responded differently if asked specifically about the social aspects.

### 4.6. Implications for Future Physical Activity and Interventions

All participants seemed to understand that physical activity is beneficial but this knowledge was insufficient to sustain physical activity levels over time. More than half reported they were less physically active than two to three years ago. The reduction was more prevalent among African Americans. The main reasons for decline in activity were health issues, lack of time, interest, or motivation. Less than 10% reported they were more active. The main reasons for sustaining or increasing activity levels were to meet personal goals, to sustain health, and pride in remaining active. Based on the findings, physical activity programs need to be oriented toward helping older adults achieve their personal goals and designed to help them sustain and improve their overall health and quality of life and should provide acknowledgement that the participants are doing a good thing for themselves. Therefore, trainers should talk with the participants to find out why they want to exercise, what they hope to achieve, and help them to focus on and remember their goals, reminding them from time to time of their stated personal goals. Trainers can also review and remind the participants of the health benefits of physical activity and acknowledge the efforts of the participants from time to time. Special efforts should be made to promote physical activity among older African Americans because they appear to have a higher rate of decline of physical activity levels over time.

### 4.7. Implications for Future Research

In-depth interviews with a larger sample better balanced in terms of gender and other important sociodemographic characteristics would allow more exploration of the effects of culture and social group. These results could also be compared with quantitative analyses of the social determinants of physical activity and changes in physical activity over time with particular attention to factors related to sustaining activity as opposed to decline. The suggestions emanating from this research and similar findings emphasizing the importance of having goals related to maintaining physical activity and strategies for overcoming pain, fatigue, and other physical and emotional barriers should be incorporated into clinical and research protocols for tailoring exercise prescriptions for individual older persons and tested in rigorously designed clinical studies, particularly in longitudinal studies that would allow measurement not only of the effectiveness in initiation of exercise but also in sustaining it. We need to test not only the ability of a particular approach to stimulate the initiation of an exercise program but also to sustaining it over time.

## 5. Conclusions

There is evidence of widespread understanding that remaining physically active is beneficial to one’s health. For many older adults, however, this knowledge alone is insufficient to sustain physical activity. It is evident both from the prevailing models of behavioral change related to physical activity and from our results that many need help to identify personal, meaningful goals as well as guidance for overcoming pain, fatigue, and other physical and emotional barriers to physical activity. To do this effectively, a person-centered, individualized approach is needed. This calls for tailoring physical activity assessment and prescription to the needs and concerns of the individual older adult.

Sustaining physical activity is at least as important as initiating physical activity, yet this has attracted far less attention in the literature on physical activity in older adults. The results of this study highlight the inadequacy of a general “exercise is good for you” message and the importance of having personal, meaningful goals for remaining active and overcoming the physical and emotional barriers noted by participants.

In conclusion, the results reported here add to and reinforce calls for a re-framing of our approach to physical activity for older adults. From encouragement of a lifetime of physical activity to individual tailoring of the physical activity assessment and prescription, there is still much to learn and to do to help older adults sustain their physical activity through the later years.

## Figures and Tables

**Figure 1 geriatrics-06-00057-f001:**
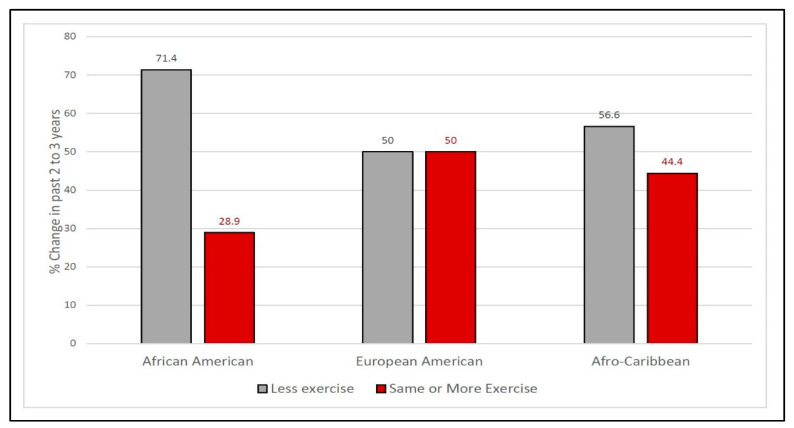
Percent reporting declined, sustained, or increased activity in the last 2–3 years by ethnic group.

**Table 1 geriatrics-06-00057-t001:** Qualitative sample characteristics by ethnic group (*n* = 62).

	African American	European American	Afro-Caribbean		
Continuous Variables	M (SD)	M (SD)	M (SD)	F	*p*
Age in years	74.90 (7.50)	76.95 (9.18)	75.76 (7.31)	0.36	0.69
Years of education	11.80 (3.04)	17.78 (4.65)	11.06 (3.23)	19.36	<0.01 *b
Years in U.S.	74.92 (7.48)	67.59 (17.45)	36.25 (15.37)	38.00	<0.01 *c
Cognition (MMSE)	26.55 (2.79)	27.58 (4.15)	27.68 (3.66)	0.60	0.55
Function (FAQ)	2.14 (3.00)	1.73 (3.95)	2.64 (4.87)	0.26	0.77
BMI kg/m^2^	43.29 (6.56)	35.22 (3.81)	37.66 (6.52)	11.00	<0.01 *a
Categorical Variables	African American	European American	Afro-Caribbean	Phi Coefficient	Fisher’s Exact Test *p*
Gender	2 (M) 19 (F)	14 (M) 10 (F)	4 (M) 13 (F)	0.45	<0.01 *b
Born in the U.S.	0 (no) 21 (yes)	4 (no) 20 (yes)	15 (no) 2 (yes)	0.78	<0.01 *c
Receiving Medicaid	15 (no) 6 (yes)	22 (no) 1 (yes)	14 (no) 3 (yes)	0.27	0.09

Both the MMSE and the FAQ scores range from 0–30; ethnic group planned comparisons statistically significant group differences (a = African American, b = European American, c = Afro-Caribbean). * Significant at *p* ≤ 0.01 level.

**Table 2 geriatrics-06-00057-t002:** Binary logistic regression of ethnic differences across physical activity trends while controlling for age and BMI (*n* = 62).

							95% CI for OR	
	B	SE	Wald	df	*p*	OR	Lower	Upper
Ethnicity (EA)	-	-	3.138	2	0.208	-	-	-
AA	1.68	1.08	2.40	1	0.122	5.35	0.64	44.67
AC	2.50	1.46	2.92	1	0.088	12.19	0.69	214.57
Age	−0.08	0.04	3.30	1	0.069	0.92	0.85	1.01
BMI	0.05	0.09	0.28	1	0.594	1.05	0.88	1.26
Constant	3.34	4.04	0.69	1	0.408	28.22	-	-

Overall model not statistically significant (*χ*2(4) = 6.42, *p* = 0.170). EA = European American, AA = African American, AC = Afro-Caribbean.

## Data Availability

The data presented in this study are available on request from the corresponding author. The data are not publicly available due to privacy concerns.

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
