# Peer review of "Sustaining or Declining Physical Activity: Reports from an Ethnically Diverse Sample of Older Adults"

_geriatrics, 2021, doi:10.3390/geriatrics6020057_

Round 1

Reviewer 1 Report

This reviewer recognizes the importance of this kind of research. It analyzes the levels of physical activity practice in different population groups of older adults. Especially when different variables of interest are assessed.

The study aimed to examine reports of maintenance or decline in physical activity levels over time and participants' explanations of these trends in a diverse community-based sample of older adults.

After reading the document I have important doubts about it. I will ask you in the following lines about it.

  • I don't really understand what kind of study have you design. There is a lack of important information about qualitative exploration. Also, I consider that there are significant limitations in the quantitative analysis. Please, try to explain it clearer.
  • Lines 31-33. There is a new revision of the World Physical Activity Recommendations (WHO, 2020). It should be indicated as a reference instead that you used at this point.
  • It would be necessary to provide data on physical activity levels previous, and the characteristics of the sample.
  • Lines 54-55. You should include a reference that can support this claim.
  • The introduction section should be reorganized in order to understand more clearly the aim of the study. It must be specified to a greater extent.
  • Line 60. What systematic review are you referring to?
  • Lines 73-75. The objective does not contemplate the option of having older adults who increase their level of physical activity.
  • The type of study carried out more precisely should be indicated. Can you explain what is mixed analysis?
  • The sample in your study is so small. Why this number of participants? Are there similar studies on which it has been based?

Please, try to explain better it.

  • How the age of the participant was taking into account in the data analysis? There is a 34-year-old difference among the participants, I consider that it can modify the results, don´t you?
  • Line 95. The characteristics of the interview are not described. Relevant information is missing.
  • How was the qualitative data analyzed? Categories, criteria, Etc.
  • How was the Mini-Mental Estate Examination conducted? It is not explained.
  • Information is missing on how the responses were encoded. What criteria have you followed?
  • No information is provided on ethical issues, informed consent, etc.
  • Table 1. Units of measure are not provided.
  • Why did you do not take into account the BMI in the data analysis as a covariate? According to the data that you have presented, all the groups analyzed have a significant degree of obesity. Therefore, BMI can be an important variable to consider because it can have an influence on the levels of physical activity practice. It is not explained clearly enough in the text.
  • There were significant differences in BMI between ethnic groups. Other variables also have significant differences since the beginning of the study. How was this aspect controlled? What criteria were followed and why?
  • Qualitative data are provided but the criteria for obtaining the data are not indicated.
  • This reviewer has significant doubts about the procedure that you use for obtaining the results. It leads to very general statements. Also, it has not been proving in some cases or described in other sections.
  • The limitations section reflects a large part of the bias of this study. In addition, I have important doubts about the methodological design it. Possibly, due to an insufficient description of the procedure.

Reviewer 2 Report

GERIATRICS-1145914 presents results from a mixed methods study related to physical activity for older adults. While some parts of this manuscript were interesting, other areas could be improved. I hope the authors consider my feedback.

MAJOR COMMENTS

  • The Introduction could use revision in many areas. Physical activity participation for health and longevity is effectively a scientific fact so there is little need to underscore. Be more to the point about the potential impact of the study purpose, while also funneling into the purpose statement.
  • Survey Data: Using Medicaid as a proxy for income is a bit of a stretch given the age of the sample. There were likely many retired persons as well.
  • Survey Data: Please list how any data were categorized.
  • Quantitative Analysis: There needs to be a lot more detail in this section. Just listing a GLM is not enough. What were the specific variables examined? What were the specifics about the GLM?
  • Table 1 does not stand-alone because we don’t know which groups are significantly different.
  • The quantitative portion of the Results text could be improved. For example, consider inserting relevant statistics from the GLM?
  • The significance of the study could be bolstered in the Discussion. How may this research transform interventions and physical activity programs for older persons? This information may parallel the future research implications.

MINOR COMMENTS

  • Introduction: The first sentence runs on a bit. Consider revising to be more pointed.
  • Line 31-32: The first part of this sentence needs a supporting citation.
  • Lines 34-35: Again, supporting citation needed.
  • Lines 39-45: Supporting citations needed for nearly all the sentences. Please consider appropriate citations for the whole document.
  • Lines 86-87: Place, “n=” in front of each frequency.
  • Table 1: F-values and Phi Coefficients are not needed because they inform p-values.
  • Table 1: The presentation of the categorical variables could be improved.
  • Figure 1 does not stand-alone. What are the abbreviations? What are the p-values alluding to? No need to list the coefficients again.
  • Make any changes to the abstract that could be aligned with those made in the text.

Round 2

Reviewer 1 Report

I agree with the changes that you did in your paper

Author Response

Thank you for your comments and suggestions for revisions.

Reviewer 2 Report

  • Please include your explanation for the Medicaid/Medicare as income in the revision letter to the appropriate location of the manuscript.
  • Lines 93-98: The formatting of this paragraph needs revision.
  • Line 116: How were age and BMI ascertained? Were there any other measures collected? Why were only age and BMI controlled?
  • The y-axis formatting on this Excel generated table needs formatting. The specific percentages could also be removed on the Figure.
